# Impact of Thinning on the Yield and Quality of *Eucalyptus grandis* Wood at Harvest Time in Uruguay

Fernando Resquin [1,*], Karen Baez [2], Sofia de Freitas [2], Diego Passarella [3], Ana Paula Coelho-Duarte [4] and Cecilia Rachid-Casnati [1]

1 National Research Program of Forest Production, National Agricultural Research Institute, INIA Tacuarembó, Route 5 km 386, Tacuarembó 45000, CP, Uruguay; crachid@inia.org.uy
2 Agronomic Engineering, School of Agronomy, University of the Republic, Av. Eugenio Garzón 780, Montevideo 12900, CP, Uruguay; karenyanet10@hotmail.com (K.B.); defreitasrivero@gmail.com (S.d.F.)
3 Forest Engineering, Northeast Regional Center, University of the Republic, Route 5 km 386, Tacuarembó 45000, CP, Uruguay; diego.passarella@cut.edu.uy
4 Forestry Department, College of Agronomy, University of the Republic, Garzón Avenue 780, Montevideo 12900, CP, Uruguay; paula.coelho@fagro.edu.uy
* Correspondence: nando@inia.org.uy

**Abstract:** Understanding how thinning strategies impact wood quality and quantity for different purposes is of interest, given that plantation management is often based on parameters that require validation under varying growth conditions. Planted forests for solid purposes in the northern region of Urugay, western Argentina and South of Brazil are usually managed in initial stockings ranging from 800 to 1200 trees·ha$^{-1}$ depending on the use of clones or seeds. Subsequent thinnings are applied (at plantation ages varying from 3 to 11 years) up to final stockings of around 200 trees·ha$^{-1}$. This study evaluated contrasting thinning regimes applied early in the crop cycle, with an initial tree density of 840 trees·ha$^{-1}$. Two thinning treatments were applied at 1.5 and 7.3 years, reducing tree densities to 700–400 and 400–100 trees·ha$^{-1}$, respectively. Growth analyses were conducted from 1.5 to 20.8 years, considering total height, diameter at breast height, individual volume, total and commercial volume per hectare, mean annual increase, and current annual increase. At the final harvest, contrasting tree densities of 100, 250, and 400 trees·ha$^{-1}$ were sampled to assess wood density and mechanical properties (bending and compression on small-scale clear samples). Individual growth and wood properties were related to a Stand Density Index to understand the effect of competition on these values. The results identified thinning regimes that resulted in the most significant individual and per-hectare growth (both in thinning and clear felling) and the optimal harvest time under specific growth conditions. We assessed the proportions of commercial logs for sawmill and pulp uses, providing valuable inputs for subsequent economic analyses of thinning regimes aiming for the most convenient combination of wood products. Wood's physical and mechanical properties were relatively little affected by contrasting levels of competition between trees; therefore, the choice of silvicultural system will depend on production and economic criteria.

**Keywords:** *Eucalyptus grandis*; rose gum; thinning; growth; wood quality; sawmill





## 1. Introduction

Management of even-aged stands involves setting production objectives according to required forest products. For many non-integrated foresters, the goal is to manage plantations to obtain multiple products of higher and lower value, such as solid wood and pulp. Silvicultural decisions, including stand density management, must consider cost/benefit relationships based on yield and wood quality, among other factors. In Uruguay, plantations typically have initial densities ranging from 1000 to 1500 trees·ha$^{-1}$ and are managed with thinning operations to reduce densities up to 100 to 300 trees·ha$^{-1}$ [1].

Reducing competition between individuals by removing smaller trees concentrates growth factors such as water, light, and nutrients in individuals with the highest productive potential [2]. Thinning forests increases individual growth rates due to enhanced photosynthesis in the middle and lower canopy areas associated with greater light penetration [3]. According to Medhurst and Beadle [4] this is particularly evident for species that are shade-intolerant, although some authors claim that eucalyptus trees tolerate competition between individuals [5,6].

Having information on how these plantations should be managed to obtain high-quality products such as straight stems, large diameters and desirable wood properties (e.g., density, strength, fiber length) is essential for silviculture-based crop planning (e.g., [7]). High-quality wood is achievable by reducing competition between individuals through stand stocking regulation, concentrating growth factors in individuals with the highest productive potential [2]. The effects on growth are more pronounced in diameter at breast height and basal area compared to individual height, although responses vary [8–12]. This effect on growth rate tends to shorten rotation length, which is financially advantageous [13]. The economic income from the wood produced in commercial thinning, which essentially has a cellulosic destination, must be considered of interest as well.

Site quality, age and size of trees, and the proportion of trees removed are significant factors affecting growth response to thinning [14–17]. Site quality, closely linked to growth rate, determines that thinning effects are site-specific [18]. In high-quality forest sites, such as those in northern Uruguay [19], dominant trees are expected to respond well to thinning, as they are more efficient than smaller trees when competing for light [18]. Considering the moment of thinning for commercial eucalypt species in general, it is observed that when this coincides with maximum growth rate (within the first five years), the most significant responses are achieved in terms of individual growth [15], considering the apparent absence of limiting factors such as water and nutrients [20]. According to Smith and Long [21] after a rapid initial growth, a decrease occurs due to a closure of the canopy, which coincides with the stage of maximum leaf area development. This occurs in the early years for many eucalypt species and represents an indicator when to apply the first thinning in order to temporarily reduce the effects of competition between individuals.

Thinning intensity, expressed in terms of basal area, wood volume, or number of trees, refers to the number of interventions and the number of trees removed in each thinning entry [22]. In most cases, very intense thinning leads to an unwanted increase in the size of the branches, which reduces the quality of the wood and a loss of volume per hectare, altering the tapering. On the other hand, very light thinning leads to growth stagnation in addition to reduced diameters of little commercial value [9]. At both extremes, there is an underutilization of site occupancy potential in each site in terms of growth factors or some limitations for their expression [23]. A site-invariant relative occupancy measure which defines the density of stocking relative to a species-specific stocking maxima, Reineke [24] proposed the stand density index (SDI) as a management tool to define stocking density relative to a species-specific maximum, providing size-density relationships to understand stand competition.

Wood properties for solid eucalypt species are greatly influenced by growth rate, which depends on silvicultural practices, site characteristics, and environmental factors [25–28]. However, some studies show no relationship between wood quality and individual growth [29]. Basic wood density shows variable relationships with growth rate, with positive, negative, and neutral associations reported [30–32]. Sapwood basic density is typically greater than heartwood density in eucalyptus trees due to increased fiber diameter and reduced lumen size with age [25,33,34]. According to those authors this is explained by the rapid increase in volume and width of the cell wall with age as a result of fiber diameter increment and lumen reduction in eucalypt species. On the other hand, a positive relationship has been determined between the proportion of sapwood and the leaf area index [35]. As mentioned above, the leaf area index is directly related to the spacing between individuals. This depends on the shading tolerance shown by the different eucalyptus species [36]. However, contrary results

were obtained by Wilkins [37] with very similar values of sapwood proportion in treatments without and with thinning at the age of 2.75 years (1089 and 479 trees·ha$^{-1}$, respectively) in *E. grandis*.

The positive relationships between spacing and wood strength in eucalypt species are explained by the effect on wood density [38] and the microfibrillar angle [39,40]. Hein et al. [41] reported a combined and additive effect of the density and the microfibrillar angle (of the S2 wall) with a better capacity to predict the resistance properties of the wood than if they are considered individually. Lower values of the microfibrillar angle result in higher compressive strength, while higher values result in lower bending strength, which indicates that an ideal value will depend on the end use of the wood [42–44]. The variation in the angle of the microfibrils is associated with the proportion of late/early wood, juvenile/adult wood, and growth rate (among others). However, this is more evident in conifers than in broadleaf trees [43]. The literature, in general, shows that high growth rates determine high values of the microfibrillar angle [28,44,45] and that an increase up to 16 degrees reduces density and MOE, while values above that have no appreciable effect [40]. This determines that the effects of thinning on the microfibril angle and the properties of the wood properties must be determined for each production system.

As a consequence of the above, we hypothesize that the different intensities of thinning affect productivity, rotation age, and wood properties in trees of harvest age. To understand the magnitude of this effect, the following objectives were proposed: (i) to evaluate the response of different thinning combinations on individual and per hectare growth along the rotation, (ii) to understand how thinning combinations affect the biologically optimum harvest age, (iii) to quantify the impact effect of thinning on commercial wood, focusing on sawn and pulpwood and (iv) evaluate the effect of contrasting stocking on physical and mechanical properties of the wood.

## 2. Materials and Methods

Our analysis is based on a thinning trial comprising 10 treatments. For analyzing the first and second objectives we used thinning records and measures taken along the life of the trial to assess diameter breast height, total height, standard deviation of diameters, mortality, individual volume, volume per hectare, mean annual increment, and current annual increment. The third objective was assessed by applying a logging simulation model using information of each treatment at harvest age as inputs to quantify solid wood and pulp wood yielded by each treatment during commercial thinning and clear felling. For objective number four, we analyzed bending strength (b) and compression strength parallel to the fibers (c) by sampling trees from 3 of the 10 treatments. We extracted discs and specimens to measure modulus of elasticity (*MOE*), modulus of rupture (*MOR*) and wood basic density (ρ) in laboratory conditions. The Stand Density Index (*SDI*) [24] was calculated for each treatment to understand the evolution of competition and its relationship with growth and wood properties. Finally, statistical comparisons of all the regimes regarding growth and wood properties were performed. Methodological details of those processes are shown below.

### 2.1. Study Area

The trial was installed in a commercial *Eucalyptus grandis* plantation (seminal origin) in December 2000 in northern Uruguay (Department of Rivera, 31°15′30″ South latitude and 55°40′44″ West longitude) (Figure 1).

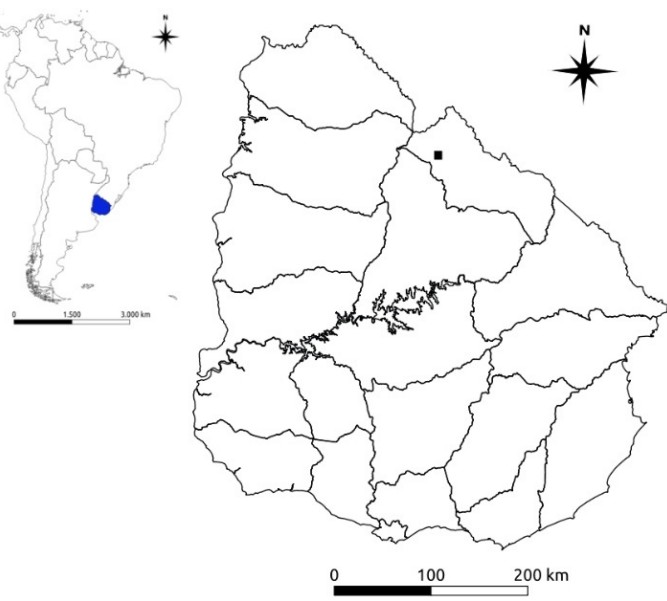

**Figure 1.** Trial location at Department of Rivera in the north of Uruguay.

Soils are reddish brown, sandy with high aluminum content, and very low natural fertility. Soil depth is greater than 1.2 m, providing a great water storage capacity [46]. Climate is temperate with average maximum and minimum temperatures of 24 and 12 °C, respectively. The average annually accumulated precipitation is 1600 mm [47]. Site index (index age = 10 years) is 31 m, which corresponds to high growth rates.

Seeds to produce the seedlings were from Bañado de Medina Experimental Sation of the School of Agronomy, University of the Republic, in the department of Cerro Largo (northeast of Uruguay). For soil preparation, a clearing furrower (with symmetrical mold-boards), three chisels, and two hilling discs were used. Planting was carried out manually with a subsequent application of 100 and 120 g per plant of ammonium phosphate to both sides of the plant.

*2.2. Experimental Design*

This experiment is part of several evaluations assessing different timing and frequencies of thinning with this eucalypt species, from which we expect to obtain information on various schemes of such interventions. There is empirical evidence that due to high growth rates, trees begin to compete at early ages, leading to a reduction in potential diameter growth. Additionally, local market conditions indicate some difficulty in selling small-diameter wood, which drives the production of large-diameter logs. This implies that initiating thinning at advanced stages of the rotation results in a reduction of individual tree growth until harvest age. The experiment comprised thinning regimes combining stocking densities during the first and second thinning applied at ages 1.5 and 7.3 years, respectively, ensuring that the remaining trees were homogeneously distributed in the plot (Table 1). The design was complete randomized blocks with 3 repetitions. Plots are composed of 8 rows of 12 plants, planted at 4 × 2 m. In total, 30 plots occupy 28,800 m ($960 \text{ m}^2$ each) with an initial stand density of 850 trees·ha$^{-1}$ (See Figure S1 in Supplementary Materials). To ensure that the information obtained in this case can be extrapolated to commercial plantations, the trial was installed under the same silvicultural management conditions (tillage type, genetic material, fertilization, weed control, etc.) used by the forestry companies in the region.

**Table 1.** Description of the thinning regime evaluated. Values represent the remanent trees per hectare for each treatment.

| First Thinning (1.5 Years) | | Second Thinning (7.3 Years) | | Rotation Age (20.8 Years) |
|---|---|---|---|---|
| Prescribed | Effective | Prescribed | Effective | Observed |
| 400 | 372 | 100 | 111 | 97 |
| 400 | 389 | 150 | 142 | 132 |
| 400 | 403 | 200 | 198 | 188 |
| 550 | 545 | 200 | 188 | 188 |
| 550 | 542 | 250 | 250 | 250 |
| 550 | 573 | 300 | 292 | 292 |
| 700 | 677 | 300 | 299 | 299 |
| 700 | 628 | 350 | 347 | 340 |
| 700 | 667 | 400 | 403 | 396 |
| No thinning | 840 | No thinning | 774 | 708 |

### 2.3. Growth and Competition

Diameter at breast height (*DBH*, cm, 1.3 m) and total height (*Ht*, m) were measured for each tree in 2002, 2005, 2007, 2008, 2009, 2011, 2015, 2018, 2019, and 2021. Total individual volume under bark (*Vi*, $m^3$.tree) was estimated applying Equation (1) [48], and the total volume per hectare (*Vht*, $m^3.ha^{-1}$) was calculated as the sum of individual volumes of each plot, related to one hectare.

$$Vi = 0.00003242 \times DBH^{1.80} \times Ht^{1.178}. \tag{1}$$

*Vht* value extracted in the second thinning was estimated by multiplying the average *Vi* by the number of thinned trees in each plot.

Coefficient of variation of the *DBH* (*CV*, %) and the number of trees per hectare (*N*, trees·$ha^{-1}$) were calculated to compare uniformity of individuals and mortality (*Mo*, %) dynamics along the rotation. Competition between individuals of each regime was assessed through Reineke's *SDI* considering the slope of the self-thinning line as −1.605 as proposed by the author. The maximum *SDI* (*SDImax*) was set on 1200 trees per hectare according to Rachid (unpublished data), and relative density (*RD*) was calculated as (*SDI/SDImax*) × 100.

To estimate commercial volume per hectare (*Vhc*, $m^3·ha^{-1}$) for pulp and sawmilling uses, a simulation software was applied. The software initializes with stand characteristics and requirements of log dimensions for each category (pulp and sawmilling) to estimate diameter distributions and stems' taper [48]. Log length considered was 5.3 m and small end diameters for pulp and sawmilling were 6 and 25 cm, respectively.

### 2.4. Wood Properties

#### 2.4.1. Weighted Basic Apparent Density

Basic density *(Bd*, g.·$cm^{-3}$) was analyzed at age 7.3 years (second thinning) and at 20.8 years. It was determined using the dry weight and green volume values of sampled stem discs according to the standard UNIT 237:2008 [49].

During the second thinning, 2 to 3 trees from each plot whose *DBH* was close to the mean of each plot were felled. Two discs were removed: one at breast height (1.3 m), and the second at 50% of the commercial height defined by a small end diameter of 6 cm over bark. Both discs were transported in closed nylon bags and placed in water on the same day to maintain saturation condition. Discs were kept submerged until a constant weight and the green volume of each piece was obtained by water displacement. Subsequently, discs were dried in an oven at 103 °C ± 2 until constant weight to obtain the dry weight.

For age 20.8 years 3 thinning regimes were selected to analyze the combination for the first and second thinning of 400–100, 550–250, and 700–400 remaining trees·$ha^{-1}$, respectively. For all plots corresponding to those treatments, 4 individuals per plot were

sampled according to *DBH* class: two trees corresponding to the mean class, one from the smaller class, and one from the higher class of each plot. A total of 36 individuals were sampled. Trees were felled and disc samples were extracted at heights of 3.3, 5.3, and 7.3 m to determined *Bd* according to the same standard.

The weighted basic density (*Bdw*, g.cm$^{-3}$) of each tree was estimated according to the following Equation (2) [50]:

$$Bdw = \frac{\sum(Ai \times Bdi)}{\sum Ai} \tag{2}$$

where *Ai* and *Bdi* are the cross-sectional surface area of the disk and wood densities of the disks extracted at 3.3, 5.3, and 7.3 m, respectively.

### 2.4.2. Current Apparent Density

Current apparent density (*Cd*, g·cm$^{-3}$), was assessed for age 21.8. From the same stems sampled for estimating *Bdw*, two logs of 1 m length located at 2.3 and 4.3 m height were extracted. Specimens from the intermediate position (50% in the radius) and from the north-south orientation were prepared of two dimensions: 25 × 25 × 100 mm and 25 × 25 × 400 mm (width, thickness, and length, respectively) (Figure 2). These samples were also used for bending and compression tests.

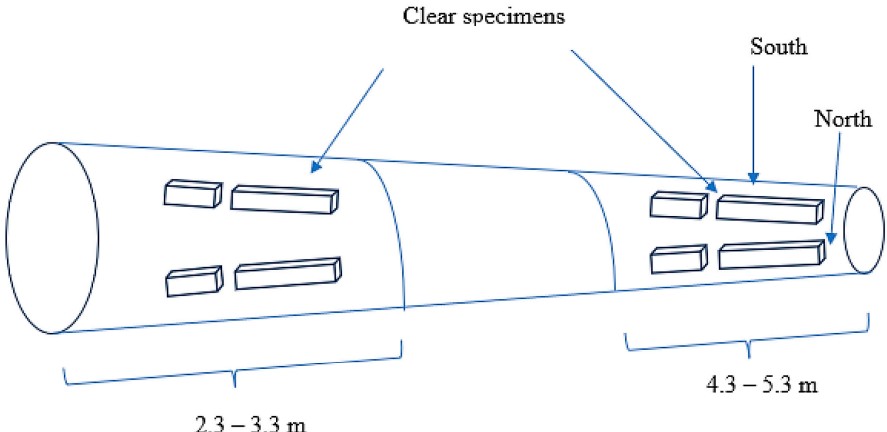

**Figure 2.** Localization of specimens within sampled logs.

The 288 specimens (2 per log × 2 logs per tree × 2 dimensions × 36 trees) were conditioned at 20 °C and 65% of humidity for 20 days until constant mass was reached. The equilibrium moisture content was close to 12% at the end of conditioning. This moisture content was verified in 63 test specimens.

Each specimen's weight and volume were measured to determine the *Cd* according to the standard UNIT 237:2008 [49]. The *Cd* value of each tree was obtained as the average of the four specimens of each log.

### 2.4.3. Bending Strength

The mechanical tests were carried out on the 25 × 25 × 400 mm specimens in a universal testing machine with a maximum load capacity of 100 KN (Shimadzu AGS-100, Kyoto, Japan) with a load cell of accuracy of 0.5%. Procedure for estimating the static bending strength followed usual standards ASTM D 143-94; UNIT 1137:2007 [51,52].

Modulus of Elasticity (*MOEb* MPa) and Modulus of Rupture (*MORb*, MPa) were calculated, both in bending according to Equations (3) and (4):

$$MOEb = \frac{\Delta PL^3}{48\Delta dI} \tag{3}$$

where: $\Delta P$ = Range of load between two points of the elastic zone (N) typically 10 to 40% of maximum load were taken, $L$ = span (mm), $\Delta d$ = Elongation (mm), and $I$ = moment of inertia (mm).

$$MORb = \frac{3PL}{2bh^2} \qquad (4)$$

where $P$ = maximum applied load (N), $L$ = span (mm), $b$ = width of the specimen (mm), and $h$ = height of the specimen (mm).

2.4.4. Compression Strength Parallel to the Fibers

Measurement of the compressive strength was carried out using the smallest specimens in the same equipment and according to the standards ASTM D 143-94; UNIT 1137:2007 [51,52]. Modulus of Elasticity (*MOEc*, MPa) and Modulus of Rupture (*MORc*, MPa) were calculated, both in compression according to Equations (5) and (6):

$$MORc = \frac{Pmax}{Ao} \qquad (5)$$

$$MOEc = \frac{*\Delta P.Lo}{\Delta L.Ao} \qquad (6)$$

$P_{max}$ is the maximum load attained during each test. $\Delta P/\Delta L$ is the slope of the load-displacement curve between 10 and 40% of the maximum load reached during the test, $Lo$ is the initial length of the specimen, and $Ao$ is its initial cross section area.

*2.5. Statistical Analysis*

To assess the effect of the different regimes on the range of variables of interest an analysis of variance was undertaken according to the following model Equation (7):

$$Yij = \mu + ei + \beta j + \varepsilon i \qquad (7)$$

where $Y_{ij}$ is the variable measured in the plot for each thinning $i$, in block $j$; μ is the overall mean of trial; $ei$ is the effect of thinning (fixed effect); $\beta j$ is the $j$ effect of the block; and $\varepsilon_{ij}$ is the experimental error associated with each observation (assumed to be independent and with a normal distribution of mean 0 and variance $\sigma^2$).

The variables analyzed were *DBH, Ht, Vi, Vht, Bbp, Cd, MOEb, MORb, MOEc, and MORc* at age of 20.8, and *Vht* and *Bdw* at the age of 7.3 years.

Normality of the residuals and homogeneity of the variances were verified with the Shapiro–Wilk and Brown–Forsythe tests, respectively, whereas the distribution of the errors was checked through graphic analysis. In cases where these assumptions were not verified, the Kruskal–Wallis test was used. Subsequently, comparisons of means were carried out using the Bonferroni and SNK tests for growth variables and Tukey tests for wood property variables.

Correlations between *Cd/Bdw* vs. *MOFb, MORb, MOEc,* and *MORc* were analyzed through the Pearson test using the values of each stem sampled.

**3. Results**

*3.1. Individual Growth*

The growth curve of *DBH* and *Ht* shows a change in the behavior of both variables from the moment of second thinning onwards, from which the differences between thinning regimes increase (Figure 3). This response is more significant compared to the first thinning at 1.3 years. The differences at rotation age (20.8) are comparatively greater in the *DBH* than in *Ht*, with maximum and minimum values at lowest and highest densities, respectively. The final stand stockings of 200 and 300 trees·ha$^{-1}$, whose initial densities were 400–550 and 550–700, respectively, show very similar growth at the end of the evaluation period. *Vi* follows *DBH* behavior due to its greater weight compared to *Ht* (Figure 3). For the individual variables, a relative stagnation in growth rates is observed after 18 years of

age. *DBH* variation (*CV*) observed differs between thinning regimes. Densities with more than 300 trees·ha$^{-1}$ showed more variation (between 70%–90% of the mean). Whereas regimes with less than these population showed less variation in *DBH*. However, the most severe thinning regime showed higher variation as well at the final age. The analysis of variance at the age of 20.8 years indicates significant differences for the three mentioned variables (Table 2). However, densities with differences of 50 trees·ha$^{-1}$ showed small variation in individual growth considering final densities of 100 to 150 trees·ha$^{-1}$. For those stockings the mean *DBH* is greater than 50 cm. On the other hand, the smallest *DBH* and *Vi* are recorded in the unthinned stand. At the same time, mean height remains similar in contrasting stockings such as 300 and 800 trees·ha$^{-1}$.

**Table 2.** Mean values (±SE) of *DBH*, *Ht* and *Vi* for all regimes at 20.8 years. Different letters indicate significant differences based on a Bonferroni test ($\alpha = 0.05$).

| Thinning Regimes (Remaining Trees·ha$^{-1}$) | *DBH* (cm) | *Ht* (m) | $V_i$ (m$^3$) |
|---|---|---|---|
| 400–100 | 52.7 (0.3) a | 49.6 (0.6) ab | 4.2 (0.12) a |
| 400–150 | 50.7 (0.7) ab | 50.4 (0.1) a | 4.0 (0.14) ab |
| 400–200 | 47.2 (0.4) b | 48.9 (0.2) ab | 3.4 (0.03) b |
| 550–200 | 46.4 (1.5) bc | 47.2 (0.4) b | 3.2 (0.18) bc |
| 550–250 | 42.7 (0.2) c | 48.0 (1.0) b | 2.8 (0.08) c |
| 550–300 | 40.9 (0.5) cd | 46 (0.5) bc | 2.6 (0.04) cd |
| 700–300 | 39.5 (0.7) cd | 46.1 (0.4) bc | 2.4 (0.12) cd |
| 700–350 | 39.8 (1.0) cd | 46.3 (1.1) bc | 2.5 (0.14) cd |
| 700–400 | 37.8 (0.5) d | 43.7 (1.0) c | 2.1 (0.08) d |
| No thinning | 30.4 (0.7) e | 39.6 (0.8) c | 1.5 (0.05) e |

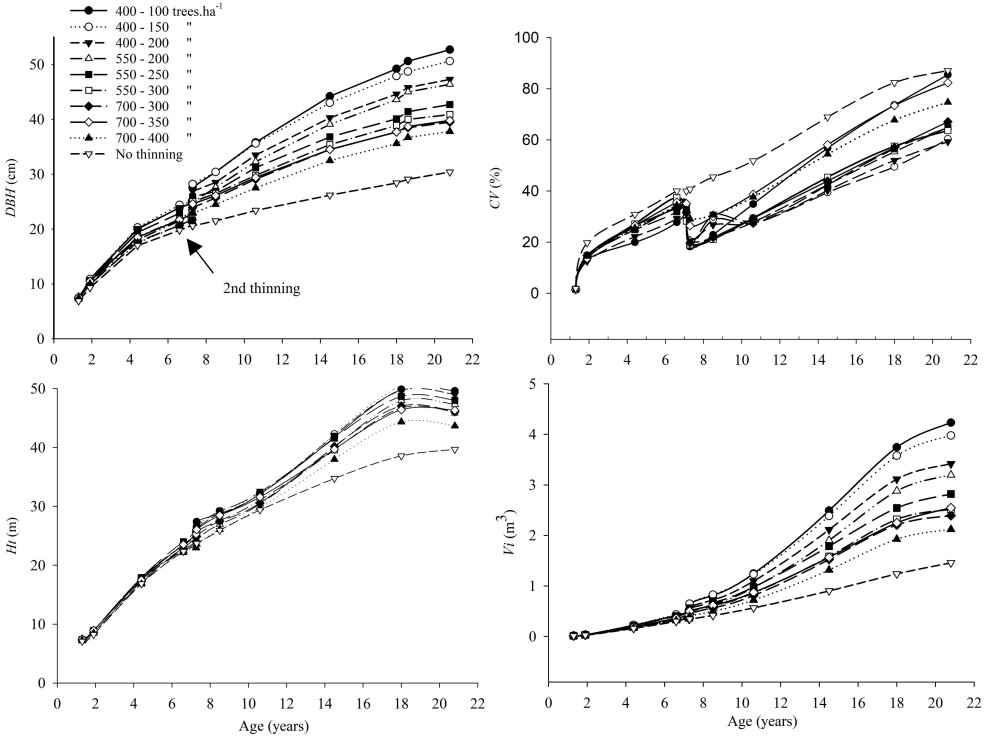

**Figure 3.** Growth of mean *DBH*, *CV*, *Ht* and *Vi* for all thinning regimes evaluated.

### 3.2. Productivity per Hectare and Competition Level

The evolution of the *Vht* shows that treatments with the largest tree densities doubles the volume of regimes with the smallest remaining tree densities. A notable decrease in the growth rate after 18 years of age is also observed for all the regimes (Figure 4). The

*Mo* showed increasing levels until the second thinning but without any relationship with the stand stockings. The reduction in the number of trees that occurred in the second thinning determines a rapid drop in the loss of trees which subsequently recovers the levels prior to this moment. In all cases, relatively low values of *Mo* are observed except for populations with 400–100 trees·ha$^{-1}$ and the control without thinning. The analysis of *MAI-CAI* curves indicates that the biologically optimal harvest age differs little between stockings and occurs around the age of 20 years (Figure 5a–c).

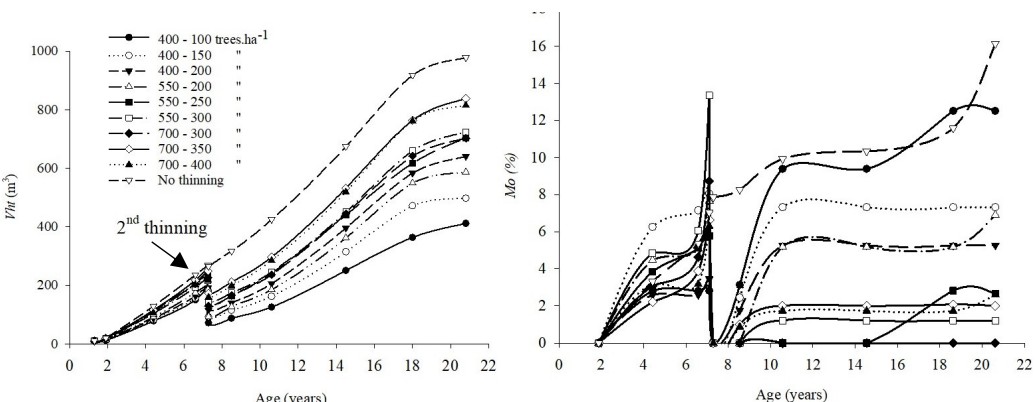

**Figure 4.** *Vht* and *Mo* for all the thinning regimes evaluated.

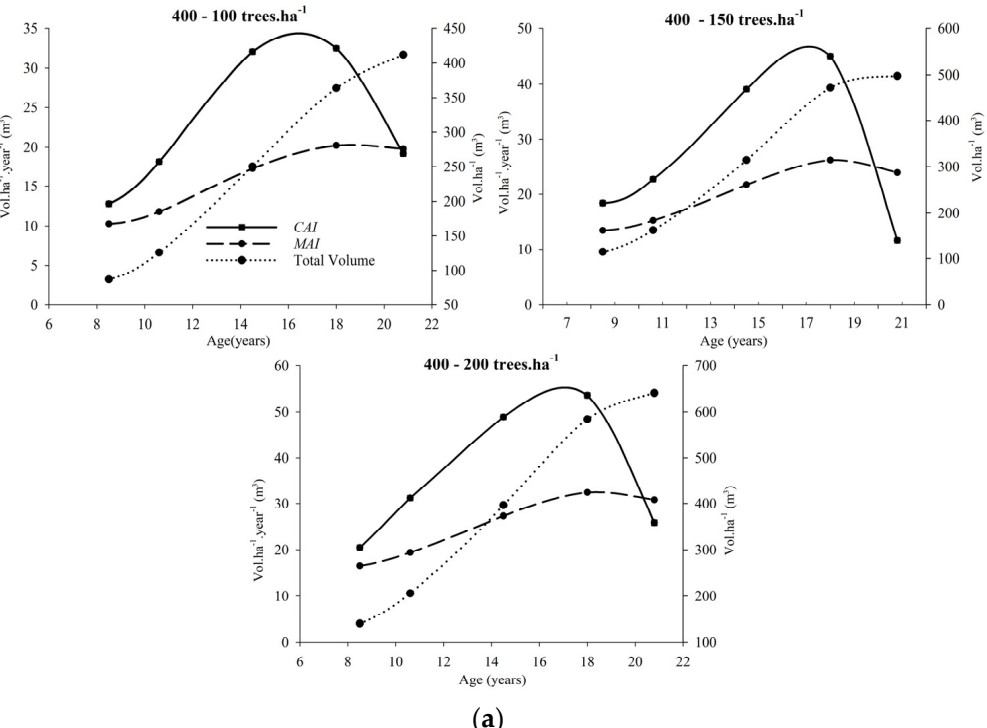

(**a**)

**Figure 5.** *Cont.*

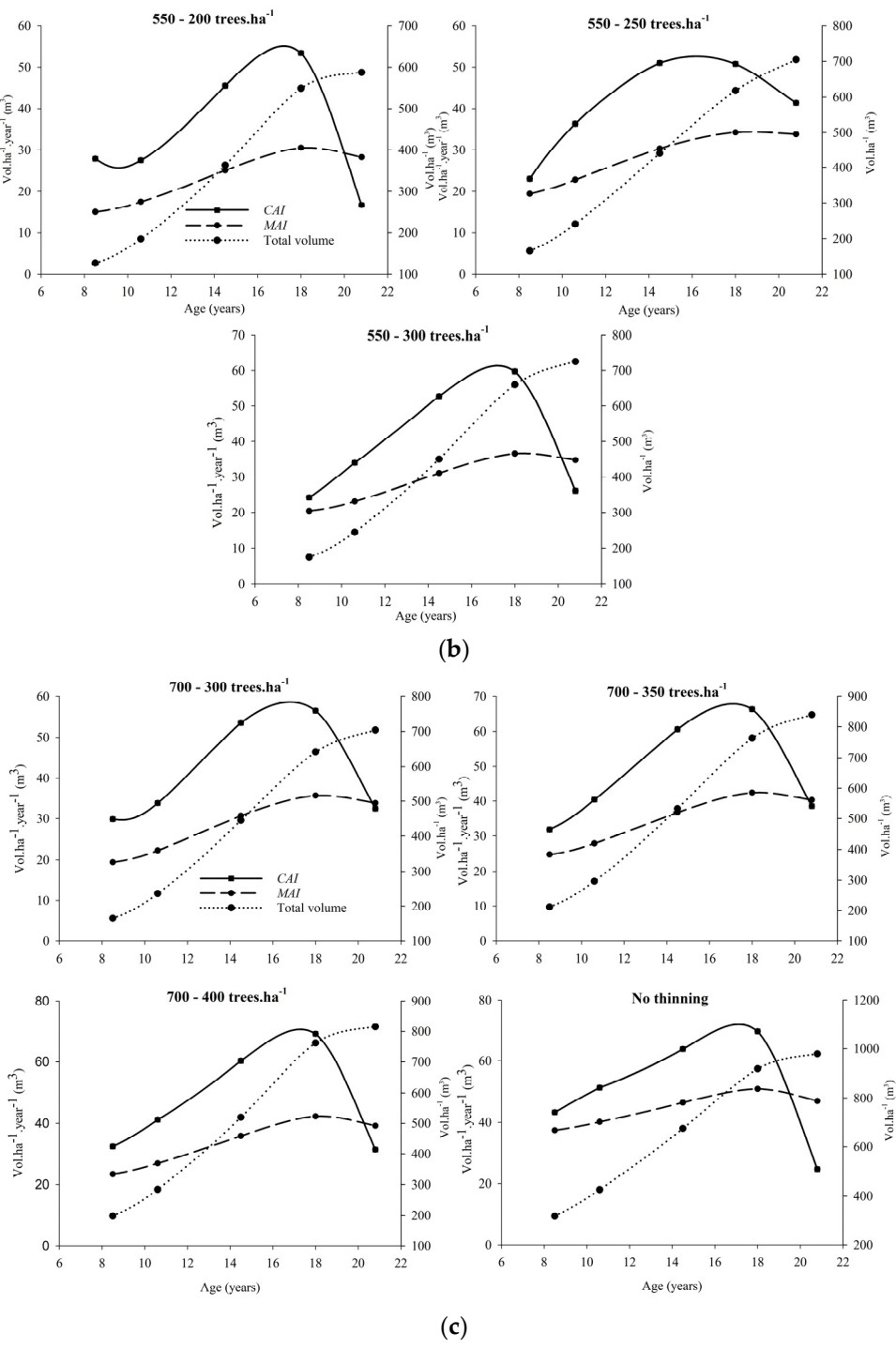

**Figure 5.** (**a**). *MAI-CAI* curves and *Vht* of thinning regimes of 400–100, 400–150, and 400–200 trees·ha$^{-1}$ until 20.8 years. (**b**). *MAI-CAI* curves and *Vht* of thinning regimes of 550–200, 550–250, and 550–300 trees·ha$^{-1}$ until 20.8 years. (**c**). *MAI-CAI* curves and *Vht* of thinning regimes 700–300, 700–350, 700–400 trees·ha$^{-1}$ and No thinning until 20.8 years.

As expected, total volumes harvested in the second thinning were the larger in regimes with most significant reduction in the number of individuals, such as the regimes 400–100, 550–250/200, and 700–300 trees·ha$^{-1}$. An inverse trend is recorded with the values obtained at harvest age and total wood harvested (Table 3). A joint plot of *Vht* and *DBH* is performed to understand the relationship between the total volume of wood, and trees fulfilling the requirement of large diameter for sawmilling industries. An inverse relationship between

both variables was observed, with final densities of approximately 250 trees·ha$^{-1}$, which would provide relatively high wood yields and diameter values (Figure 6).

**Table 3.** Mean values (±SE) of *Vht* extracted and remaining after thinnings. Different letters indicate significant differences between regimes (alpha = 0.05) based on a Bonferroni test.

| Thinning Regimes (Trees·ha$^{-1}$) | 2nd Thinning (7.3 Years) | Rotation Age (20.8 Years) | Total |
|---|---|---|---|
| | -------------------------(m$^3$.ha$^{-1}$)------------------------- | | |
| 400–100 | 113 (2.1) a | 412 (35.3) d | 525 (33.6) d |
| 400–150 | 88 (3.2) b | 498 (35.8) cd | 586 (38.4) cd |
| 400–200 | 75 (3.3) bc | 641 (23.4) c | 716 (26.6) c |
| 550–200 | 97 (4.4) ab | 587 (30.3) c | 684 (34.7) c |
| 550–250 | 103 (5.6) ab | 704 (19.7) bc | 807 (19.9) b |
| 550–300 | 71 (5.9) bc | 724 (15.2) bc | 795 (9.6) b |
| 700–300 | 109 (13.6) ab | 703 (24.3) bc | 812 (19.9) b |
| 700–350 | 77 (8.5) bc | 839 (41.4) ab | 916 (34.7) a |
| 700–400 | 60(5.9) c | 815 (19.7) b | 875 (18.2) ab |
| No thinning | - | 978 (24.0) a | 978 (24.0) a |

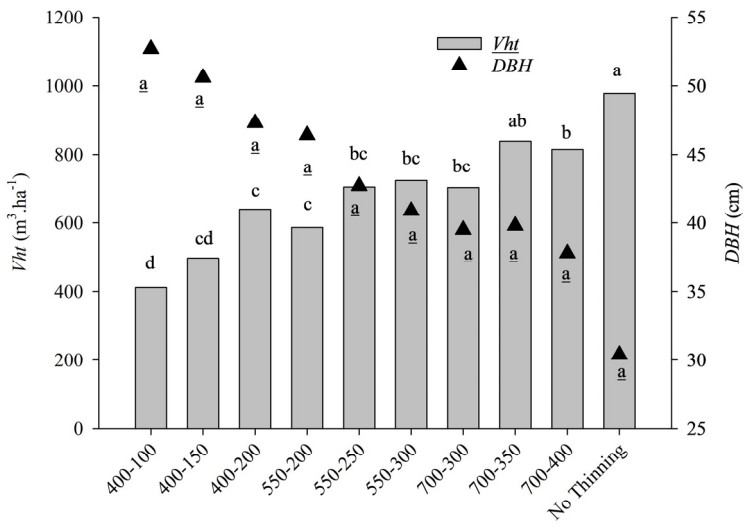

**Figure 6.** Mean values of *DBH* and *Vht* for the different stand stockings of remaining trees at 20.8 years. Different letters indicate significant differences between thinning regimes at alpha = 0.05 based on a Bonferroni test.

Regarding competition, the first intervention was performed when the population was theoretically in free growth, suggesting it was an early thinning. At the age of the second thinning, two regimes with the lowest tree densities were still in free growth, while the remaining regimes were in the stage corresponding to increasing competition. None of the regimes were fully stocked. At harvest age (20.8 years), only the regime 400–100 trees·ha$^{-1}$ remained in theoretical free growth, the least severe regimes denoted fully stocked populations whereas the intermediate regimes were in increasing competition (Table 4).

**Table 4.** *RD* (considered as a percentage of maximum stand density index) at different ages for all thinning regimes.

| Thinning Regimes (Trees·ha$^{-1}$) | Age (Years) | | | | | | | | | | | |
|---|---|---|---|---|---|---|---|---|---|---|---|---|
| | 1.3 [b] | 1.3 [a] | 1.9 | 4.4 | 6.6 | 7.1 [b] | 7.3 [a] | 8.5 | 10.6 | 14.5 | 18 | 20.8 |
| 400–100 | 10 | 9 | 8 | 21 | 28 | 29 | 11 | 12 | 15 | 21 | 24 | 27 |
| 400–150 | 9 | 9 | 8 | 21 | 29 | 30 | 14 | 16 | 19 | 26 | 31 | 34 |
| 400–200 | 9 | 9 | 9 | 23 | 30 | 32 | 18 | 20 | 25 | 33 | 39 | 43 |
| 550–200 | 9 | 9 | 11 | 26 | 34 | 35 | 16 | 19 | 24 | 32 | 38 | 42 |
| 550–250 | 9 | 9 | 11 | 29 | 37 | 39 | 21 | 23 | 29 | 38 | 43 | 48 |
| 550–300 | 10 | 9 | 12 | 28 | 36 | 37 | 23 | 26 | 31 | 41 | 48 | 53 |
| 700–300 | 9 | 9 | 13 | 31 | 40 | 41 | 22 | 25 | 31 | 40 | 47 | 51 |
| 700–350 | 9 | 9 | 13 | 31 | 40 | 42 | 28 | 30 | 36 | 48 | 55 | 61 |
| 700–400 | 0 | 0 | 0 | 30 | 40 | 41 | 29 | 32 | 38 | 50 | 59 | 64 |
| No thinning | 9 | 9 | 15 | 37 | 46 | 48 | 48 | 51 | 58 | 70 | 80 | 84 |

Note: White: Free growth (RD < 30%) light grey increasing competition (30% < RD < 55%, dark grey: fully stocked (RD > 55%), [b]: before thinning, [a]: after thinning.

Regarding to the *Vhc* for the second thinning, for all thinning regimes log dimensions fulfilled requirements only for pulpwood dimensions. (Table 5). The proportion of logs for cellulose and sawmill at harvest age is variable among thinning combinations. Regimes with final stand densities of 100 to 300 trees·ha$^{-1}$ provided over 70% of wood for sawmilling. In the case of the treatment No thinning this proportion was close to 50%. Considering the total commercial volume (including thinning and harvest) very similar yields of wood for sawmilling are recorded comparing final densities of 100 and 200 trees·ha$^{-1}$ (close to 70%).

**Table 5.** *Vhc* for the second thinning and harvest age, categorized by final destination (sawmill and pulp), obtained using a logging simulator for the species.

| Thinning Regimes (Trees·ha$^{-1}$) | 2nd Thinning (7.3 Years) | | | Rotation Age (20.8 Years) | | | Total | | |
|---|---|---|---|---|---|---|---|---|---|
| | Sawmill | Pulp | Total | Sawmill | Pulp | Total | Sawmill | Pulp | Total |
| | ------------------------------------------(m$^3$·ha$^{-1}$)------------------------------------------ | | | | | | | | |
| 400–100 | - | 110 | 110 | 310 (88%) | 43 (12%) | 353 | 310 (67%) | 153 (33%) | 463 |
| 400–150 | - | 87 | 87 | 368 (86%) | 59 (14%) | 427 | 368 (72%) | 146 (28%) | 513 |
| 400–200 | - | 72 | 72 | 436 (83%) | 91 (17%) | 527 | 436 (73%) | 163 (27%) | 609 |
| 550–200 | - | 90 | 90 | 414 (82%) | 93 (18%) | 507 | 414 (69%) | 183 (31%) | 597 |
| 550–250 | - | 94 | 94 | 432 (76%) | 135 (24%) | 567 | 432 (65%) | 229 (35%) | 661 |
| 550–300 | - | 71 | 71 | 439 (72%) | 167 (28%) | 606 | 439 (65%) | 238 (35%) | 677 |
| 700–300 | - | 120 | 120 | 386 (69%) | 175 (31%) | 561 | 386 (58%) | 277 (42%) | 663 |
| 700–350 | - | 72 | 72 | 469 (71%) | 196 (29%) | 665 | 469 (64%) | 268 (36%) | 737 |
| 700–400 | - | 56 | 56 | 457 (66%) | 239 (34%) | 696 | 457 (61%) | 295 (39%) | 752 |
| No thinning | - | - | - | 373 (47%) | 413 (53%) | 786 | 373 (47%) | 413 (53%) | 786 |

### 3.3. Wood Properties

3.3.1. Basic Apparent Density and Current Apparent Density

The analysis of the *Bdw* and *Cd* variance shows significant differences between final densities of 100, 250, and 400 trees·ha$^{-1}$ (Figure 7). Combination of thinning 400–100 trees·ha$^{-1}$, recorded larger *Bdw* and *Cd*. The greatest variability of the data is observed for this thinning, probably associated with the greatest number of specimens extracted from different positions in the logs.

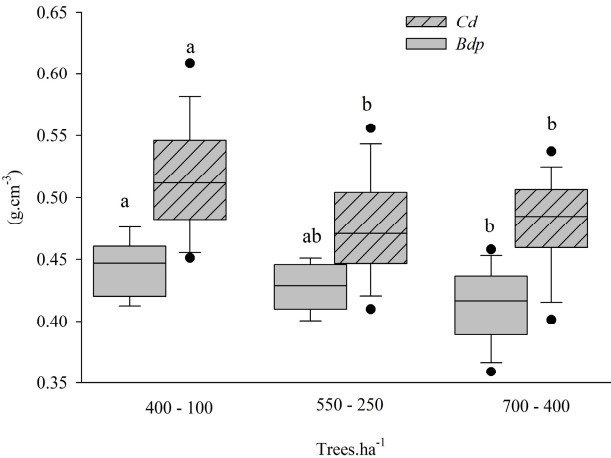

**Figure 7.** *Bdw* and *Cd* values (5th/95th percentile) of regimes of 400–100, 550–250, and 700–400 trees·ha$^{-1}$ at the age of 20.8 years. Different letters indicate significant differences between thinning regimes based on a Tukey test ($\alpha$ = 0.05).

The increase in *Bdw* from the second thinning to harvest age was similar in magnitude among lower final stand stockings (400–100 and 550–250 trees·ha$^{-1}$), whereas *Bdw* remained unchanged for higher stockings (700–400 trees·ha$^{-1}$) throughout the analyzed period (7.3 to. 20.8 years of age) (Figure 8).

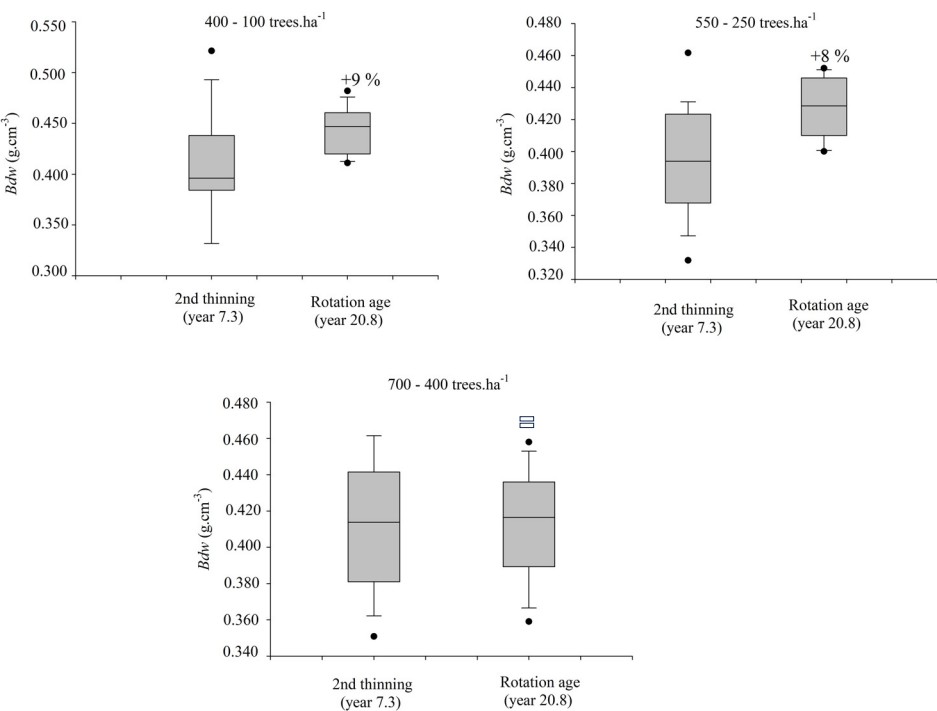

**Figure 8.** Comparison of *Bdw* values (5th/95th percentile) of the thinning regimes 400–100, 550–250, and 700–400 trees·ha$^{-1}$ at the ages of the second thinning and rotation age.



### 3.3.2. Mechanical Properties

The analysis of variance did not detect differences in the bending and compression strength between different thinning regimes (Figures 9 and 10). However, there is a tendency towards higher values, particularly with the *MOEb* and *MOEc* to the thinning combination 400–100 trees·ha$^{-1}$.

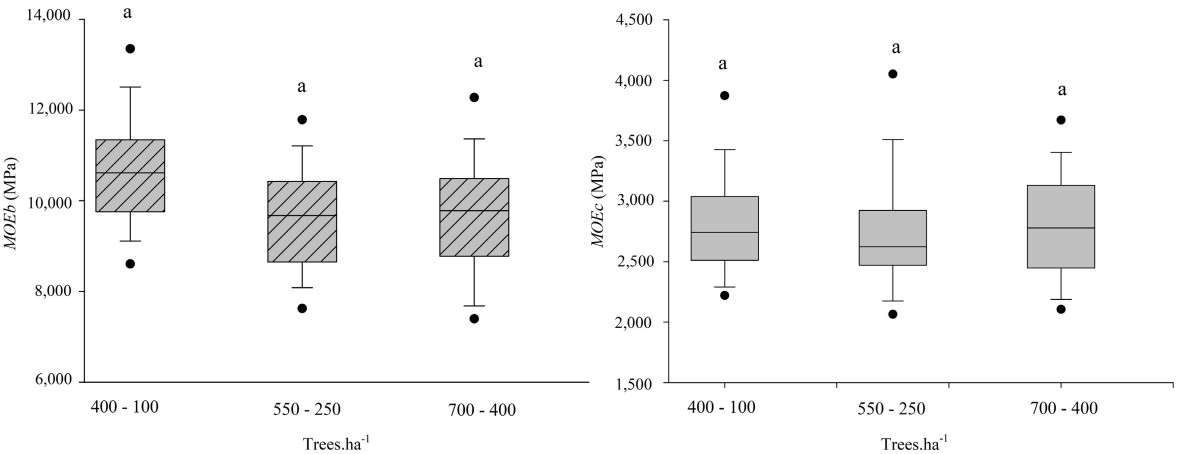

**Figure 9.** Comparison of *MOEb* and *MOEc* (5th/95th percentile) of the thinning combinations of 400–100, 550–250, and 700–400 trees·ha$^{-1}$ at the age of 20.8 years. Different letters indicate significant differences between thinning regimes based on a Tukey test ($\alpha$ = 0.05).

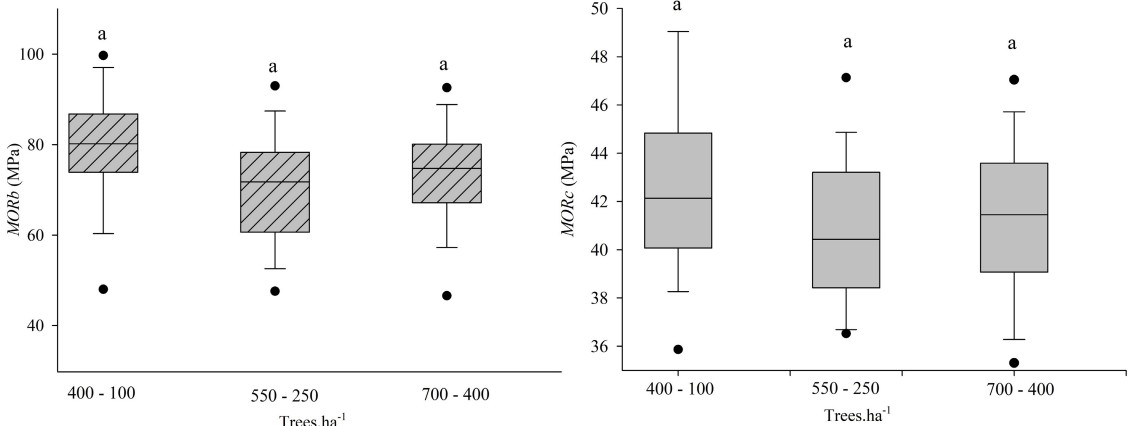

**Figure 10.** Comparison of *MORb* and *MORc* (5th/95th percentile) of the thinning combinations 400–100, 550–250, and 700–400 trees·ha$^{-1}$ at the age of 20.8 years. Different letters indicate significant differences between thinning regimes based on a Tukey test ($\alpha$ = 0.05).

### 3.4. Relationship between Wood Density and Mechanical Properties

The correlation values between *Cd* and *Bdw* and the mechanical properties of the wood indicate that the highest values (and in all cases significant) are obtained with the first of the mentioned densities except for *MOEc* (Table 6 and Figures S2–S4 in Supplementary Materials). The lower values obtained with the *Cd* are explained since all the variables were measured in the identical specimens, unlike the *Bdw* measured in the discs. A strong correlation value was also obtained between the density of the wood measured in test pieces and in discs despite coming from different positions in the log.

**Table 6.** Simple correlation values of wood density vs. mechanical resistance properties.

|  | MOEbf | MOEc | MORb | MORc | Bdw | Cd |
|---|---|---|---|---|---|---|
| Cd | 0.64 | 0.32 * | 0.64 | 0.63 * | 0.69 | - |
| p-value | <0.0001 | 0.0001 | <0.0001 | <0.0001 | <0.0001 | |
| Bdw | 0.61 | 0.27 | 0.44 | 0.57 | - | - |
| p-value | <0.0001 | 0.12 | 0.008 | 0.0003 | | |

* Values calculated with the Spearman test.

## 4. Discussion

Our work was oriented to understand growth and wood properties of *Eucalyptus grandis* growing in excellent site conditions (SI = 31 m at 10 years) at different thinning regimes, that combine a wide range of stockings. The analysis was extended to the lifespan of the stand, allowing us to visualize the optimal harvest age considering total and commercial volume (for sawmilling and pulp), along with wood quality for solid uses. To relate our results to the state of competition of the stands, we used the *SDI* as applied by Reineke [24]. Results confirm the hypothesis that contrasting thinning combinations affect the growth and some properties of the wood of *E. grandis*. Implementing thinning regimes on a commercial scale will be subject to costs and projections of market prices for each type of product, particularly for the diameters of the basal logs. Therefore, it is necessary to identify regimes that combine individual growth and yield per hectare in search of the best economic result for each particular commercial scenario.

### 4.1. Individual Growth

The first thinning was performed at the age of 1.3 years when *SDI* values indicate the lack of competition between individuals. However, competition started at age 4 for regimes with 700 trees·ha$^{-1}$, and two years later for regimes with 400 to 550 trees·ha$^{-1}$ so that at the time of the second thinning, different levels of competition provided a first differentiation in *DBH* but not in *Ht*. Prior to the second thinning, contrasting stand stockings of 850 and 400 trees·ha$^{-1}$ presented differences of 4.4 cm. After the second thinning, the large span thinning intensities applied provided a range of competition levels so that at harvest age the difference in the average *DBH* values between densities of 700 opposed to 100 trees·ha$^{-1}$ was over 22 cm. A larger *DBH* at the end of the cycle obtained with lower stockings has been reported by several authors [10,53,54]. Ferraz Filho et al. [55] observed that the higher diameters were achieved with final densities of 150 trees·ha$^{-1}$ (at age 11 years) and when the first thinning was applied sooner (at age 2.5 year) for the same species. However, in all treatments, a change in the growth rate was observed from year 11 onwards.

The analysis shows that with tree densities of 100 and 200 trees·ha$^{-1}$ *DBH* values greater than 45 cm can be achieved. Comparing different paths to final densities of 200 trees·ha$^{-1}$, populations of 550 and 400 trees·ha$^{-1}$ prior to thinning did not result in differences regarding individual diameters at harvest age. This range of final densities has also been evaluated for eucalyptus species, seeking to identify thinning combinations that allow high individual growth rates [11].

In our analysis we checked the small response of dominant trees to thinning due to a more significant relationship with site quality than with silvicultural practices. This determines that no statistical differences regarding *DBHdom* were detected between final stand stocking of 100 and 400 trees·ha$^{-1}$ (52.7 and 45.7, respectively). There is evidence that these trees have greater efficiency in capturing light and are little affected by competition with neighbors [56,57]. In this way, it is possible to obtain a certain number of trees with large diameters of trees growing at a wide range of densities [18,58].

Growth of *Ht* prior to the second thinning shows a difference of 2.2 m between contrasting densities (800 and 400 trees·ha$^{-1}$), representing a minor variation compared to *DBH*. The reduced variation in the different thinning regimes was reported by Schönau [59]. Smaller spacings promote the occurrence of dominated trees, which determines a reduction in their size [60]. However, in some cases, an inverse relationship occurs, explained by the

high competition for light [61]. The latter probably occurs with closer spacings than in the ones analyzed in this work.

As opposed to the *DBH*, growth rate decreases from year 18 onwards for all treatments except for the treatment without thinning, which begins sooner. This is probably due to a greater competition for resources. From this age onwards, a certain reduction occurs, which may be due to stagnation in growth in *Ht* added to the error range in the measurement equipment for this variable since no tree death occurred. Lack of differences in *Ht* for final stand stockings between 100 to 350 trees·ha$^{-1}$ are consistent with the results of other authors evaluating similar thinning regimes for eucalypt species [11,12].

Behavior of *Vi* is very similar to *DBH* in terms of the differential response in time after the second thinning. However, the difference between the extreme densities is almost three times (1.5 vs. 4.2 m$^3$.tree$^{-1}$). These differences are due to more significant growth in diameter and height that occurs at larger spacings. Similar relationships were obtained by Scheeren et al. [23] and Trevisan [62] evaluating eucalypt species with final trees densities equivalent to this study's.

Considering the contrasting competition status at final age indicated by the *SDI* between No thinned populations (810 trees·ha$^{-1}$, fully stocked and imminent mortality) and heavily thinned treatments (100 trees·ha$^{-1}$, in transition between free growth and increasing competition), the differences recorded in individual growth are one to three times in the thinned regimes relative to the unthinned regimes.

### 4.2. Productivity per Hectare

Due to the different reductions in stand stocking in the first and second thinning, *Vht* differs substantially between treatments. As expected, the populations with the greater stocking register the greater volumes. A similar response was obtained by Trevisan [62] evaluating *E. grandis* at 18 years of age with final densities of 800 to 200 trees·ha$^{-1}$. The results obtained in this study show that increases in tree densities do not translate in the same proportion to productivity per hectare, as reported in the literature [53,63]. In this case, stand stockings 7 times higher (700 vs. 100 trees·ha$^{-1}$) result in *Vht* increasing by almost 2.5 times (978 vs. 412 m$^3$.ha$^{-1}$, respectively).

For all thinning regimes a decrease in the growth rate is observed from 18 years onwards, indicating the proximity of the optimal harvest age. Considering *Vht*, this occurs at an age close to 20–21 years. The effect of competition between individuals is evident since the lowest tree densities produce the most significant individual growth, but this competition did not affect the harvest age, as no clear trends were observed throughout tree densities. The relationship between spacing and optimal harvest age has been evaluated by several authors [63–65] observing varying responses, indicating the complexity of the association between silvicultural variables such as site quality or genetics.

The volume extracted in the second thinning depends on the reductions in the number of trees (400–100, 550–200, and 700–300 trees·ha$^{-1}$). On the other hand, these levels do not affect the values obtained in clear felling, showing the latter's dependence on tree densities. The volumes obtained with both harvests are most directly related to final stand stockings.

The thinning regime 550–250 trees·ha$^{-1}$ maintains relatively high volume per hectare combined with individual growth in diameter. This is a relevant aspect when seeking to produce large volumes of wood and large diameters. Either for local markets or export, little end diameters larger than 30 cm over bark are required and attract better prices.

All wood obtained in the second thinning has a cellulosic destination, which has a low economic return due to the relatively low price of the wood and the cost of freight to the wood processing plants. The highest proportions of sawn wood in clear felling are obtained with the final densities of 100 and 250 trees·ha$^{-1}$ (greater than 75%) at the same time that the proportion of pulpable wood is one-third for the higher densities. Actually, this type of wood shows small variation ranging from 33 to 39%. When considering volumes of pulpable and sawn wood together, differences between the thinning regimes are minor with respect to considering pulpable wood solely. This was also reported by Larocca et al. [66]

who evaluated different thinning regimes in *E.grandis* in an evaluation under similar soil and climate characteristics to those of our trial.

Mortality is low in general. However, it was larger and accumulated throughout the lifespan of the stand up to 18.6 years for no-thinning treatment (16%). This was due to competition as the *SDI* denoted increasing competition from age 4 years and fully stocking from age 11. Higer mortality in higher stocking was also observed for the same species at a younger age by Ferraz Filho et al. [55]. However, the treatment with more severe extractions 400–100 trees·ha$^{-1}$ showed high mortality in the first years after the second thinning (12%) and is probably related to the sudden exposure of trees to wind. Lower mortality was observed in the treatments where competition was controlled through mild thinning and final stand stockings were larger than 250 trees·ha$^{-1}$.

Considering variation of diameters, regimes with higher stand stockings (700–350, 700–400 trees·ha$^{-1}$, and No thinning) had higher *CV* (over 80%) which is mostly due to crown differentiation promoted by competition. However, the most drastic thinning (400–100 trees·ha$^{-1}$) also seemed to propitiate diameter variation due to increases in maximum diameters when trees that died occurred for excessive exposure after the second thinning. Although treatments 400–150 and 400–200 trees·ha$^{-1}$ had some mortality after the second thinning, it was milder (60%) and those treatments showed the smaller variations since larger extractions of small trees during thinning compared to other treatments, lead to a more severe homogenization of diameters. Finally, treatments with final stand stockings ranging from 200 to 300 trees·ha$^{-1}$ did not differ much from the previous group (65%).

Although our analysis did not include different thinning times, it does provide valuable information on how the dynamics of a range of tree densities influence the volume and quality of wood for different production objectives. Future research should deepen into dynamics of competition and assess different thinning ages. A broader set of products and economic analysis can be included.

### 4.3. Wood Density

The greater *Bdw* obtained with the most intense thinning was verified by various authors evaluating different spacings in *E. grandis* [25,67] and with other eucalyptus species [68,69]. The relationship between stand productivity and wood density has also been evaluated, obtaining a negative relationship between both factors [70], which agrees with our work.

The wood density is explained by three parameters: the thickness of the fiber wall, the proportion of this wall in relation to all tissues, and the proportion of latewood [71]. Studies with eucalyptus species indicate that the higher density at wider spacings can be explained by a more rapid formation of adult wood [25], a higher content of latewood [72], and the sapwood [73] for the smallest ones. The higher wood density values in the thinning regime of 400–100 trees·ha$^{-1}$ is a positive aspect, considering that the most significant individual growth is obtained with these tree densities. However, the increase of 16 and 44 g.cm$^{-3}$ (in the basic and current density values, respectively) has little impact on the mechanical properties. This will be analyzed in the next point. At the same time, it must be considered that final stand stocking close to 250 trees·ha$^{-1}$ allowed significant individual and per-hectare growth. This thinning regime could be recommended to foresters that aim to produce quality wood for sawmilling.

It is common to find studies in the literature that show that wood density increases with age in several eucalyptus species. This is explained by alterations in the activity of the cambium that, among other effects, determines the transition from juvenile to adult wood and translates into alterations in anatomical parameters. We did not confirm any differences between ages (second thinning and harvest) with respect to wood density. However, differences were observed for the higher spacings compared to high stockings, indicating that the lower competition between trees would be the main factor responsible for the changes observed in wood density between the second thinning and the clear cut. The similarity in the level of *Bdw* increase with the most intense thinning regimes would

also be another element in favor of considering the final stand stocking of 250 trees·ha$^{-1}$ as a management regime to be used commercially.

### 4.4. Mechanical Properties of Wood

The tendency to obtain a slight superior wood strength with larger spacings could be associated with the positive relationship between the growth rate and the microfibrillar angle for eucalyptus species [45,74]. Wood density is also involved in the wood strength, and associated with the thickness of the wall, as analyzed previously. This weak relationship between strength and spacing was also reported by Cueto et al. [75] and Lima and García [76] when evaluating *E. grandis* of ages and thinning regimes similar to this work.

It is important to note that the values obtained with small specimens overestimate those recorded with commercial-size beams due to the absence of visible defects. According to O'Neill et al. [34,77,78] the *MOE* and *MOR* obtained in this test are 12 to 15% and 40 to 86% higher than those obtained in full-size beams. From the point of view of the categorization of wood for structural use, it is observed that the values obtained would have different suitability for such uses depending on the standard considered. Both the national use standard UNIT [79] and the regional standard ABNT [80] take as reference an average *MOE* value of 12,000 MPa for this species. On the other hand, the JAS (Japan Agricultural Standard) standard considers a minimum value of 7850 MPa for the use of wood for structural purposes [77].

Beyond these categories of use, the data obtained show that a final stand stocking of 100 trees·ha$^{-1}$ does not translate into higher quality logs from considering their technological properties. This is another element to consider in implementing less severe thinning. However, the final decision must also consider the prices and economic benefits of wood obtained.

### 4.5. Relationships between Wood Properties

The correlations between *Bdw, Cd MOE,* and *MOR* are moderate, as reported by several authors evaluating eucalyptus species [81]. However Lima and Garcia [76] determined a high correlation between density and compressive strength in a study with 18-year-old *E. grandis*. In our study, it is probably that the higher individual growth rates obtained with the most intense thinning regimes did not determine changes in the microfibril angle, which ultimately led to similar values of *MOE* and *MOR* in the thinning regimes evaluated. Therefore, the density of the wood would be the only main factor explaining these resistance values. The reduced capacity of wood density to predict *MOE* and *MOR* has been verified by Hein et al. [41].

## 5. Conclusions

The results obtained through a wide range of thinning regimes show that it is possible to obtain different productivity levels individually and per hectare. The regime to be implemented will depend on each particular situation's price relationships and production objectives, but the final stand stocking close to 250 trees·ha$^{-1}$ combines relatively high levels of productivity, low mortality and controlled diameter variation. However, it is interesting to note that with all thinning regimes, it is possible to obtain similar levels of individual growth of the dominant trees, which generally are those of the most significant commercial value.

The growth curves of the different thinning regimes evaluated allow the identification of the age of 20–21 years as the optimal harvest period from a biological point of view. Another positive aspect to highlight is that obtaining large diameters does not imply a reduction in the quality of the logs destined for sawing or peeling for veneer production. Therefore, the decision of the silvicultural system to implement at a commercial level will depend on the volume set as an objective and its economic result. Our analysis provided valuable insights of the relationship between population dynamics of *Eucalyptus grandis* and silvicultural criteria, logs destination, and wood characteristics. This information

would help to understand possible outcomes on the process regarding silviculture to be applied for desired mix of wood products.

**Supplementary Materials:** The following supporting information can be downloaded at https://www.mdpi.com/article/10.3390/f15050810/s1, Figure S1. Diagram of plots distribution and the description of the number of trees per hectare in the first and second thinning of each plot, Figure S2. Scatter plots between *Cd* values versus mechanical parameters for all thinning regimes evaluated, Figure S3. Scatter plots between *Bdw* values versus mechanical parameters for all thinning regimes evaluated, Figure S4. Scatter plot between the values of *Bdw* and *Cd* for all the thinning regimes evaluated.

**Author Contributions:** F.R., C.R.-C. and D.P. planned and designed the research, conducted fieldwork and performed experiments. K.B. and S.d.F. contributed to fieldwork, data elaboration and analysis through their degree thesis work. F.R., K.B., S.d.F., A.P.C.-D., C.R.-C. and D.P. wrote the manuscript. All authors have read and agreed to the published version of the manuscript.

**Funding:** This study was funded by the National Research Institute of Agricultural Research (INIA) through different research projects executed from 2002 to date. Currently, this area of research is being funded by the i-FORES project (INIA-FO_35_0_00: Herramientas de información integradas para el manejo silvicultural eficiente y sustentable).

**Data Availability Statement:** The data presented in this study are available on request.

**Acknowledgments:** The authors thank INIA for contributing with funding, and the LUMIN SA company for their collaborating with the field experiments and plantation. Special thanks go to the researchers Ricardo Methol for designing and initiating the experiment activities and Juan Pedro Posse for his collaboration in it and support throughout the experiment lifetime. Authors acknowledge Juliana Ivanchenko, Pablo Nuñez, Federico Rodriguez, Wilfredo Gonzalez, and Sebastián Inthamoussu for their valuable help on thinnings and trial preservation.

**Conflicts of Interest:** The authors declare no conflicts of interest.

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
