# Peer review of "Impact of Thinning on the Yield and Quality of Eucalyptus grandis Wood at Harvest Time in Uruguay"

_forests, doi:10.3390/f15050810_

Round 1

Reviewer 1 Report

Comments and Suggestions for Authors

Line 32, A more precise definition of “barely affected” based on descriptive or formal statistical analyses would strengthen the abstract.

Abstract, Were the planting densities, thinning regimes, genotype selection or any other aspect of management related to a particular management or production objective or do these represent standard practices?  More context is needed to internationalize this work.

Line 54  What is meant by “for each situation”?

Line 55-56. A better term is needed to replace “tree populations”. Perhaps “stand stocking” or “tree size-stand density relationships”?

Line 96-97 Is the sapwood-heartwood density relationship similar to the poor quality of juvenile wood described in the managed pine literature/ It would be useful to contextualize this phenomenon in Eucalyptus accordingly.  These terms are mentioned in line 111 but should perhaps be introduced here

Line 119 consider using a term different from “harvest period”  Is this the same as rotation age or  harvest age or is the timing of commercial thinning also covered in this concept?

Methods

Please make clear the rationalization for beginning thinning treatments at such an early stage of stand development.  While competition may be commencing, is there any operational value/product expectation from this treatment when trees are so small or is it merely intended to serve as a baseline? Are the spacings used standard and. If not, what is standard?  When does thinning normally commence in these systems?  Such information may help others working with different species determine the suitability of this research methods to address similar questions.

Line 430  Substitute “small” for “little”

Line 497  There appear to be  words missing between “in” and “however”

Lines 509-510  This sentence is unclear

Comments on the Quality of English Language

 English grammar is generally correct to the extent that the information is fully understandable.  However, numerous and very minor stylistic changes would be simple and  beneficial improvements to readability. I would not consider these as impacting acceptability of the manuscript as is. Minor additions to the introduction and methods are outlined in my comments to the author

Author Response

Reviewer 1

Specific comments:

Line 32, A more precise definition of “barely affected” based on descriptive or formal statistical analyses would strengthen the abstract.

The text has been modified according to the editor´s reviewer.

Line 36: …were relatively little affected by contrasting levels……

Abstract, Were the planting densities, thinning regimes, genotype selection or any other aspect of management related to a particular management or production objective or do these represent standard practices?  More context is needed to internationalize this work.

The text has been modified including a new paragraph according to the editor´s reviewer.

Lines 18-22. Planted forests for solid purposes in the northern region of Urugay, western Argentina and South of Brazil are usually managed in initial stockings ranging from 800 to 1200 trees.ha-1 depending on the use of clones or seeds. Subsequent thinnings are applied (at plantation ages varying from 3 to 11 years) up to final stockings of around 200 trees.ha-1.

Line 54  What is meant by “for each situation”?

The text has been modified according to the editor´s reviewer.

Lines 58-59: (e.g. density, strength, fiber length) is essential for silviculture-based crop planning (e.g., [7]). 

Line 55-56. A better term is needed to replace “tree populations”. Perhaps “stand stocking” or “tree size-stand density relationships”?

All the text has been modified replacing that term according to the editor´s reviewer .

Line 96-97 Is the sapwood-heartwood density relationship similar to the poor quality of juvenile wood described in the managed pine literature/ It would be useful to contextualize this phenomenon in Eucalyptus accordingly.  These terms are mentioned in line 111 but should perhaps be introduced here

The text has been modified including a new paragraph according to the editor´s reviewer.

Lines 101-103: According to those authors this is explained by the rapid increase of volume and width of the cell wall with age as a result of fibre diameter increment and lumen reduction in eucalypt species.

Line 119 consider using a term different from “harvest period”  Is this the same as rotation age or  harvest age or is the timing of commercial thinning also covered in this concept?

The text has been modified according to the editor´s reviewer.

Line 126: ….affect productivity, rotation age, and…..

Methods

Please make clear the rationalization for beginning thinning treatments at such an early stage of stand development.  While competition may be commencing, is there any operational value/product expectation from this treatment when trees are so small or is it merely intended to serve as a baseline? Are the spacings used standard and. If not, what is standard?  When does thinning normally commence in these systems?  Such information may help others working with different species determine the suitability of this research methods to address similar questions.

The text has been modified including a new paragraph according to the editor´s reviewer.

Lines: 167-177 and 180-183

Line 430  Substitute “small” for “little”

The text has been modified according to the editor´s reviewer.

Line 452: …we checked the small response…

Line 497: There appear to be  words missing between “in” and “however”

The text has been modified according to the editor´s reviewer.

Line 524: Mortality is low in general, however…

Lines 509-510  This sentence is unclear

The text has been rewritten following the reviewer´s recommendations.

Lines 533-535: Considering variation of diameters, regimes with higher stand stoking populations (700-350, 700-400 trees.ha-1, and No thinning) had higher CV (over 80%) which is mostly due to crown differentiation promoted by competition.

English grammar is generally correct to the extent that the information is fully understandable.  However, numerous and very minor stylistic changes would be simple and  beneficial improvements to readability. I would not consider these as impacting acceptability of the manuscript as is. Minor additions to the introduction and methods are outlined in my comments to the author

We appreciate the comments from the reviewer, and we hope this new version of the manuscript reaches her/his expectations. The manuscript has been completely reviewed and rewritten, looking to achieve a clearer presentation of results and highlighting the main contribution of this work.

Reviewer 2 Report

Comments and Suggestions for Authors

The authors focus on a scientifically and practically interesting problematics, namely how the thinning strategies impacts on Eucalyptus grandis wood quality and quantity. The article is the output of several projects financed by INIA, which were implemented in the last 22 years and have, in a way, at least a locally strategic silvicultural importance. The article is a suitable contribution to the given section of the journal Forests. But the meaning for the reader is, in my opinion, rather local, because no revolutionary outputs or extraordinary conclusions were reached.

I have nothing to criticize the article in its form, at most a few missing pieces of information, that could have contributed to its greater importance. At the same time, I note that the view below is more that of a woodworker than a forester, although with a certain perspective.

So, I have only a few minor comments on an article and then more personal considerations and observations.

I would add the title of the article to "... —A Case Study from the Uruguay".

In key words instead of Eucalyptus grandis put either Rose gum or Red grandis.

Think about the statement on lines 95-96. At least, it is necessary to specify what density (type) it is.

Also, assess the meaning of the sentence in lines 108-110, which has no logical explanation with respect to bending strength. Higher values of the microfibrillar angle (in S2) in this case are clearly related to the presence of juvenile wood, and therefore mostly lower density and MOR (MOE) in general.

It is a pity that the microfibrillar angle values are not part of this article, if they were determined, at least the average values at the time of the thinning and at the end of the experiments. This would be very helpful in understanding, explaining, and analysing some of the results.

And maybe it's also a pity that you didn't measure the density using a densitometer in profile along the cross-section of the trunk on the manipulated discs.

The effort to use this type of wood for sawmill processing is quite interesting, even though its primary use in the case of plantation cultivation is more in the pulp and paper industry than, for example, for construction (building) purposes.

Author Response

Reviewer 2

General comments

The authors focus on a scientifically and practically interesting problematics, namely how the thinning strategies impacts on Eucalyptus grandis wood quality and quantity. The article is the output of several projects financed by INIA, which were implemented in the last 22 years and have, in a way, at least a locally strategic silvicultural importance. The article is a suitable contribution to the given section of the journal Forests. But the meaning for the reader is, in my opinion, rather local, because no revolutionary outputs or extraordinary conclusions were reached.

I have nothing to criticize the article in its form, at most a few missing pieces of information, that could have contributed to its greater importance. At the same time, I note that the view below is more that of a woodworker than a forester, although with a certain perspective.

So, I have only a few minor comments on an article and then more personal considerations and observations.

We appreciate the reviewer's comments which will be useful for future research projects.

Specific comments:

I would add the title of the article to "... —A Case Study from the Uruguay".

The title was modified as suggested by the reviewer. We understand that it is not a case study because the experiment was carried out under the same conditions as the commercial plantations which have similar characteristics in the countries of the region (Southern Brazil and the western Argentina).

Title: Impact of Thinning on the Yield and Quality of Eucalyptus grandis Wood at Harvest Time in Uruguay

In key words instead of Eucalyptus grandis put either Rose gum or Red grandis.

The keywords were included according to the reviewer´s recommendation.

Keywords: Eucalyptus grandis; Rose gum; thinning; growth; wood quality; sawmill

Think about the statement on lines 95-96. At least, it is necessary to specify what density (type) it is.

The text has been modified according to the editor´s recommendation.

Lines 99-100: Sapwood basic density is typically greater than heartwood density …

Also, assess the meaning of the sentence in lines 108-110, which has no logical explanation with respect to bending strength. Higher values of the microfibrillar angle (in S2) in this case are clearly related to the presence of juvenile wood, and therefore mostly lower density and MOR (MOE) in general.

We agree with the reviewer on the concept that a greater microfibril angle determines a lower MOE. Therefore we modify the text in the following way:… , while higher values result in lower bending strength …(Line 115).

It is a pity that the microfibrillar angle values are not part of this article, if they were determined, at least the average values at the time of the thinning and at the end of the experiments. This would be very helpful in understanding, explaining, and analysing some of the results.

And maybe it's also a pity that you didn't measure the density using a densitometer in profile along the cross-section of the trunk on the manipulated discs.

The effort to use this type of wood for sawmill processing is quite interesting, even though its primary use in the case of plantation cultivation is more in the pulp and paper industry than, for example, for construction (building) purposes.

We agree with the reviewer that the variables suggested by the reviewer are interesting for the purposes of explaining some results. In any case, the focus of our research proposes to generate information on the effect of management on the growth and properties of wood that can be used by wood producers for solid uses.